

# Influence of exercise duration on cardiorespiratory responses, energy cost and tissue oxygenation within a 6 hour treadmill run

Hugo A. Kerhervé, Scott McLean, Karen Birkenhead, David Parr and Colin Solomon

School of Health and Sport Sciences, University of the Sunshine Coast, Sippy Downs, QLD, Australia

## ABSTRACT

**Purpose**. The physiological mechanisms for alterations in oxygen utilization ($\dot{V}O_2$) and the energy cost of running ($C_r$) during prolonged running are not completely understood, and could be linked with alterations in muscle and cerebral tissue oxygenation.

**Methods**. Eight trained ultramarathon runners (three women; mean ± SD; age 37 ± 7 yr; maximum $\dot{V}O_2$ 60 ± 15 mL min$^{-1}$ kg$^{-1}$) completed a 6 hr treadmill run (6TR), which consisted of four modules, including periods of moderate (3 min at 10 km h$^{-1}$, 10-CR) and heavy exercise intensities (6 min at 70% of maximum $\dot{V}O_2$, HILL), separated by three, 100 min periods of self-paced running (SP). We measured $\dot{V}O_2$, minute ventilation ($\dot{V}_E$), ventilatory efficiency ($\dot{V}_E$:$\dot{V}O_2$), respiratory exchange ratio (RER), $C_r$, muscle and cerebral tissue saturation index (TSI) during the modules, and heart rate (HR) and perceived exertion (RPE) during the modules and SP.

**Results**. Participants ran 58.3 ± 10.5 km during 6TR. Speed decreased and HR and RPE increased during SP. Across the modules, HR and $\dot{V}O_2$ increased (10-CR), and RER decreased (10-CR and HILL). There were no significant changes in $\dot{V}_E$, $\dot{V}_E$:$\dot{V}O_2$, $C_r$, TSI and RPE across the modules.

**Conclusions**. In the context of positive pacing (decreasing speed), increased cardiac drift and perceived exertion over the 6TR, we observed increased RER and increased HR at moderate and heavy exercise intensity, increased $\dot{V}O_2$ at moderate intensity, and no effect of exercise duration on ventilatory efficiency, energy cost of running and tissue oxygenation.

Corresponding author
Hugo A. Kerhervé,
hkerherv@usc.edu.au

## INTRODUCTION

Maximal oxygen utilization ($\dot{V}O_{2MAX}$), the fractional rate of $\dot{V}O_{2MAX}$ (% $\dot{V}O_{2MAX}$) and the energy cost of running ($C_r$), i.e., the submaximal metabolic demand per unit distance, are primary predictors of performance in marathon and ultramarathon running (*Di Prampero et al., 1986*; *Lazzer et al., 2012*), and may be affected by prolonged duration exercise. Increased $\dot{V}O_2$ and $C_r$ were measured after one hour of running at ∼80% $\dot{V}O_{2MAX}$ (*Hunter & Smith, 2007*), after 90 min at 65% and 80% $\dot{V}O_{2MAX}$ (*Xu & Montgomery, 1995*), and after

self-paced marathon events (*Hausswirth et al., 1996*; *Kyröläinen et al., 2000*; *Nicol, Komi & Marconnet, 1991*). It has been suggested that increases in $C_r$ correlates positively with distance covered (*Brueckner et al., 1991*; *Lazzer et al., 2012*). However, for longer durations of running, this relationship was not observed: $\dot{V}O_2$ and $C_r$ increased only during the first 8 h of a 24-h treadmill run performed at ~40% of the velocity associated with $\dot{V}O_{2MAX}$ (*Gimenez et al., 2013*), and there was no change in $C_r$ despite increased energy cost of cycling after a 65 km trail ultramarathons, (*Millet et al., 2000*). Also, following a 65 km mountain ultramarathon, $C_r$ did not change and $\dot{V}O_2$ increased when tested on level ground (10 km h$^{-1}$), $C_r$ and $\dot{V}O_2$ did not change when tested on uphill gradients (10 km h$^{-1}$, 5% grade), but $C_r$ and $\dot{V}O_2$ increased on a downhill gradient (10 km h$^{-1}$, $-5$% grade) (*Vernillo et al., 2015*). The situation changed again following a 330 km mountain ultramarathon, where $C_r$ (at 6 and 8 km h$^{-1}$) and the energy cost of walking (at 5 km h$^{-1}$) improved (decreased) on an uphill gradient (15%) (*Vernillo et al., 2016b*; *Vernillo et al., 2014*).

The physiological basis for increased $\dot{V}O_2$ and $C_r$ could be related to decreases in cardiorespiratory efficiency, such as increased heart rate (HR) (*Kyröläinen et al., 2000*), increased pulmonary ventilation ($\dot{V}_E$) and decreased pulmonary efficiency ($\dot{V}_E{:}\dot{V}O_2$) (*Millet et al., 2000*; *Vernillo et al., 2016b*; *Vernillo et al., 2014*). The different outcomes in $\dot{V}O_2$ and $C_r$ responses observed following ultramarathons could be attributed to decreased respiratory exchange ratio (RER) (*Gimenez et al., 2013*; *Kyröläinen et al., 2000*; *Vernillo et al., 2016b*; *Vernillo et al., 2015*; *Vernillo et al., 2014*). However, other tissue-specific adaptations could also contribute to variations in $\dot{V}O_2$ and $C_r$ during prolonged running, such as increased neural input and impairments in the structural integrity of the musculoskeletal system (*Kyröläinen et al., 2000*; *Vernillo et al., 2015*). For instance, tissue oxygenation in the vastus lateralis muscle (VL) measured using near-infrared spectroscopy (NIRS) decreased following a 330 km running ultramarathon, without changes in $\dot{V}O_2$ (*Vernillo et al., 2016a*), though the specific effects of the structural disruption of muscle tissue and of exercise duration on tissue oxygenation were not tested. Contrasting with these results, VL oxygenation increased following a 45-min, high intensity intermittent running exercise (*Sear et al., 2010*), VL muscle oxygen utilization and muscle blood flow increased following ~95 min of trail running (*Vercruyssen et al., 2012*), and VL oxygenation did not change during 4-h of cycling including three, 80 min exercise periods at 45% of maximal power output (*Rupp et al., 2013*). Therefore, the specific effects of exercise duration on tissue oxygenation are not known.

Oxygenation of the prefrontal cortex (PFC) could also be directly involved in exercise regulation by integrating signals of homeostatic disturbances, and decreases in PFC oxygenation could reflect a decreased exercise tolerance (*Robertson & Marino, 2016*). Although cerebral tissue oxygenation did not change as a function of duration during 4-h of cycling (*Rupp et al., 2013*), exercise intensity may have been too low for comparisons with existing studies in prolonged running except for those conducted during 24 h or 300 km ultramarathons, and no research has currently described PFC oxygenation during prolonged running. Indirect estimations of neural input can also be gained using ratings of perceived exertion (RPE) (*DeMorree, Klein & Marcora, 2012*), which have been described to correlate with HR and ventilatory rate (*Hampson et al., 2001*; *Mihevic, 1981*;

*Nicolò, Marcora & Sacchetti, 2015*; *Tucker & Noakes, 2009*), but not $\dot{V}_E$ or $\dot{V}O_2$ (*Hampson et al., 2001*; *Nicolò, Marcora & Sacchetti, 2015*).

Therefore, the aim of this study was to determine muscle and cerebral tissue oxygenation, cardiorespiratory ($\dot{V}O_2$, $\dot{V}_E$, $\dot{V}_E:\dot{V}O_2$, respiratory exchange ratio [RER], HR), $C_r$ and RPE within a 6 h running exercise on a treadmill (6TR). The duration of exercise was selected for comparison with previous literature (*Millet et al., 2000*; *Vernillo et al., 2015*), including a study having measured decreased PFC activity (*Wollseiffen et al., 2016*), and to correspond to the acute phase of alterations in $\dot{V}O_2$ and $C_r$ observed in a previous study (*Gimenez et al., 2013*). To establish the specific effect of exercise duration, we aimed to measure the selected parameters at a moderate intensity (*Gaesser & Poole, 1996*) corresponding to the expected average speed of a group of trained ultramarathon runners, and at a heavy exercise intensity (*Gaesser & Poole, 1996*) minimizing the contribution of eccentric contractions to mechanical work, i.e., uphill (*Minetti, Ardigo & Saibene, 1994*). Based on the specific literature, we hypothesized that: (1) cardiorespiratory and RPE responses would increase throughout the 6TR; (2) tissue oxygenation would decrease as a function of duration only in PFC at the moderate intensity, and decrease in PFC and muscle tissues at heavy intensity; and (3) $C_r$ responses would remain unchanged throughout the 6TR.

## METHODS

### Ethical statement and participants

This study was approved by the Human Research Ethics Committee of the University of the Sunshine Coast (Approval Number: S/12/432). Participants were informed of the experiment protocol and associated risks, and provided written informed consent for participation. The participant group consisted of eight healthy and trained ultramarathon runners (three women and five men: mean ± SD; age: 37.4 ± 7.4 yr; height: 176 ± 5.3 cm; weight: 70.5 ± 8.9 kg; $\dot{V}O_{2MAX}$: 60.1 ± 14.5 mL kg$^{-1}$ min$^{-1}$).

### Study procedures
#### Design

The participants attended the laboratory for two separate sessions within a 15-day period. At the first session, participants completed screening procedures (medical history questionnaire, pulmonary function, anthropometric characteristics) and an incremental maximal running test on a treadmill (T200, Cosmed, Rome, Italy) to volitional exhaustion, consisting of 3 min at 6 km h$^{-1}$ and 0% grade, then 1 min stages incremented by 2.5% grade each min at 6 km h$^{-1}$ up to a maximum of 10% grade, then 3 min stages incremented by 1 km h$^{-1}$. For the running test, $\dot{V}O_2$ was measured continuously using a two-way, non-rebreathing valve (series 2700: Hans-Rudolph, Kansas City, KS, USA) and an automated open-circuit spirometry respiratory gas analysis system (True One 2400, Parvo Medics, Sandy UT). We monitored HR using a chest strap and watch (RS800; Polar Electro, Kempele, Finland; Ambit, Suunto Oy, Vantaa, Finland) recording at 1 Hz. The incremental maximal running test was used to determine maximum heart rate (HR$_{MAX}$), $\dot{V}O_{2MAX}$, and the velocity eliciting 70% of $\dot{V}O_{2MAX}$ (70% $v\dot{V}O_{2MAX}$) at 10% grade.

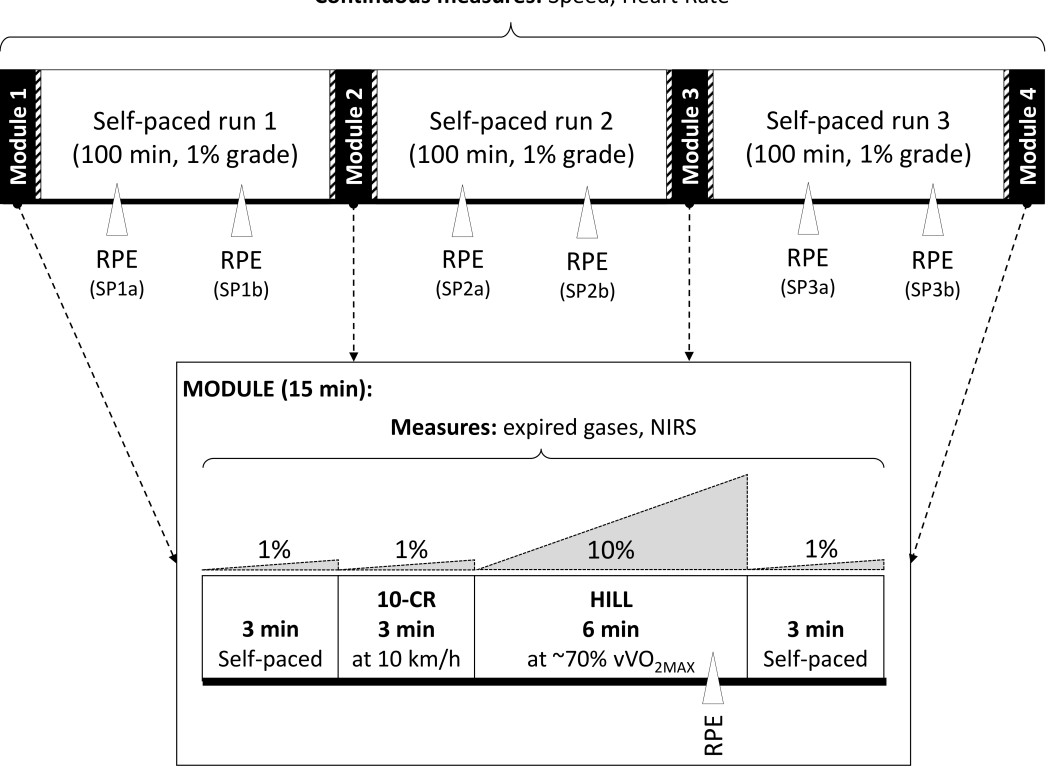

**Figure 1  Six hour treadmill run protocol.** Schematic representation of the timing of exercise and measurements performed during the 6 h treadmill run (6TR).

At the second session, participants completed the 6TR with the aim to run the maximum distance possible, which included four repetitions of a 15 min testing module at 0, 105, 225 and 345 min, each separated by 100 min of self-paced running (SP) (Fig. 1). Each module consisted of 3 min at a self-selected speed, 3 min of moderate intensity running at a fixed speed of 10 km h$^{-1}$ (10-CR), 6 min of high intensity running at the speed eliciting 70% v$\dot{V}O_{2MAX}$ at a 10% grade (HILL), and 3 min at a self-selected speed. For HILL, the participants had the option to decrease the speed if they perceived they would be unable to complete the condition and/or complete the 6TR, and were instructed to select the highest possible speed for the condition. This modification had to happen during the first two minutes of HILL to ensure data was collected at steady state. One participant completed the modules at 0, 110, 235 and 345 min, but the difference in timing was considered to have no effect on the combined results. During the SP periods participants could vary their speed, walk or stop *ad libitum* (time was not stopped for toilet breaks, and the distance to the toilets was added to the total), and were provided with performance feedback if requested. Participants were instructed to follow their normal nutrition and hydration routine. The actual running duration was 6 h, and the mean total time stopped (to attach and remove the testing equipment for the modules) was 18 ± 1 min (Fig. 1).

### Performance and relative exercise intensity

Running time, speed, distance, gradient and HR were monitored continuously throughout the 6TR (the treadmill was calibrated for speed and gradient using standard procedures). The ratio of heart rate and speed was used to calculate the Physiological Cost Index (PCI = HR/V, where HR is the average heart rate in beat min$^{-1}$, and V is the running speed in m min$^{-1}$; PCI is expressed in beat m$^{-1}$). This index is a better indicator of cardiovascular drift and metabolic efficiency than HR only, as it includes the effect of speed on physiological load (*Gimenez et al., 2013*). Speed, %HR$_{MAX}$ and PCI were reported for the three SP periods (average of SP1, SP2, SP3) and during the modules at steady state (30 s average during the final minute of 10-CR and HILL conditions).

### Gas exchange measurements

The rate of oxygen utilization ($\dot{V}O_2$), the rate of carbon dioxide production ($\dot{V}CO_2$), minute ventilation ($\dot{V}_E$), ventilatory equivalent ($\dot{V}_E{:}\dot{V}O_2$) and the respiratory exchange ratio (RER = $\dot{V}CO_2{:}\dot{V}O_2$) were measured during the modules of the 6TR using the same apparatus as for the first session, which was calibrated shortly before data collection. Temperature, humidity and barometric pressure were monitored throughout the tests, and the system was recalibrated if any changes were observed. Sampling lines were removed and left to dry between modules. Absolute rate (g min$^{-1}$) of fat oxidation (OX$_{FAT}$) was calculated based on the assumption that the excretion of urinary nitrogen was negligible using Eq. (1) (*Frayn, 1983*):

$$OX_{FAT} = 1.67 \times \dot{V}O_2 - 1.67 \times \dot{V}CO_2 \tag{1}$$

### Energy cost of running

We calculated the energy cost of running ($C_r$, in mL kg$^{-1}$ m$^{-1}$) as the ratio of net $\dot{V}O_2$ (mL min$^{-1}$ kg$^{-1}$) to speed (m min$^{-1}$) at steady state (30 s average during the last minute of 10-CR and HILL). As $C_r$ is influenced by the progressive shift to $\beta$-oxidation (RER drift) independent of speed during prolonged exercise such as ultramarathon running (*Gimenez et al., 2013*), we also calculated the energy cost of running expressed as energy expenditure per unit distance ($C_r^J$, in J kg$^{-1}$ m$^{-1}$) using an energy equivalent of oxygen ranging from 19.6 to 21.1 depending on RER. An increase in $C_r^J$ indicates a decrease in running efficiency.

### Muscle and cerebral tissue oxygenation

To assess muscle and cerebral tissue oxygenation, we used 2-wavelength continuous wave NIRS devices (Portamon and Portalite, Artinis Medical Systems BV, Zetten, The Netherlands), measuring changes in deoxyhemoglobin ($\Delta[HHb]$) and oxyhemoglobin concentrations ($\Delta[O_2Hb]$) from an arbitrary baseline, and an index of tissue oxygenation (TSI = $\Delta[O_2HB]/\Delta[HHb] \times 100$). The devices were placed on the VL (at the center of the muscle belly), the lateral head of the Gastrocnemius (Gn, at the center of the muscle belly), and on the prefrontal cortex (PFC, placed above the eyebrow, between the midline of the skull and the temporalis muscle, corresponding to the region between Fp1 and F3 of the modified international EEG 10–20 system, and adjusted individually depending on

the largest response during rhythmic and sustained contractions of the upper and lower limbs). The position of each device was marked on the skin to ensure repeatability of placement. Each device was secured to the skin using adhesive elastic tape, and shielded from ambient light using $20 \times 20$ cm opaque cloth. The NIRS devices were attached and removed immediately before and after each module.

Adipose tissue thickness was measured at the two muscle sites using Harpenden calipers (British Indicators Ltd, Burgess Hill, UK) as the thickness influences the NIRS signal (*Van Beekvelt et al., 2001*). As all participants had adipose tissue thicknesses less than half of the inter-optode distance (17.5 mm, to ensure an adequate depth of illumination), no participants were excluded (VL: $4.10 \pm 1.99$ mm, range 1.6–7.6 mm ; Gn: $4.76 \pm 3.06$ mm, range 1.5–8.5 mm). Using the recommendations of the manufacturer, we used a fixed differential pathway factor (DPF) of 4 for all participants for the muscle sites, and an age-dependent DPF was calculated for the PFC (*Duncan et al., 1996*). The devices collected data at 10 Hz and the TSI was averaged over 30 s during the final minute (from 20 to 50 s of the minute, to exclude potential effects associated with transition phases) of 10-CR and HILL to obtain data representative of the exercise state. Attaching and removing the devices before and after each module was approximately 5 and 1 min, respectively, and was deducted from total running time.

### Perceived exertion

The 6–20 linear Borg scale was used to measure perceived exertion dissociated between general RPE ($RPE_{GEN}$), the conscious effort excluding any somatic sensation ($RPE_{CON}$), and the local (muscular) RPE of the knee extensor muscles excluding any psychological contribution to exertion ($RPE_{KE}$). The scale included written descriptors, and participants were familiarized with the scale during the first session. To minimize the confounding effects of the modules, the variables pertaining to self-paced exercise were recorded 33 min from the start, and 33 min before the end of each SP bout (at ∼48, 82, 163, 196, 278 and 312 min of total elapsed time during the 6TR) which are referred to as SP1a, SP1b, SP2a, SP2b, SP3a and SP3b, respectively.

## Statistical analyses

Data are reported as mean $\pm$ SD and expressed in percentage change from self-paced period (SP1) or module (M1). All data were initially tested for normality (Shapiro–Wilk's test). For the data measured during the self-paced periods of running, a one-way, repeated-measures (RM) ANOVA was used to assess the effect of duration (SP1, SP2, SP3) on speed, $\%HR_{MAX}$, PCI, and a two-way RM ANOVA was used to assess the effect of duration (SP1a, SP1b, SP2a, SP2b, SP3a, SP3b), category ($RPE_{GEN}$, $RPE_{CON}$, $RPE_{KE}$) and the interaction (duration $\times$ category) in RPE variables. For the variables collected during the modules, a one-way, RM ANOVA was used to assess the effect of duration (M1, M2, M3, M4) on $\dot{V}O_2$, $\dot{V}_E$, $\dot{V}_E{:}\dot{V}O_2$, RER, $OX_{FAT}$, $C_r$, $C_r{}^J$ and $\%HR_{MAX}$; a two-way, RM ANOVA was used to assess the effect of duration, category ($RPE_{GEN}$, $RPE_{CON}$, $RPE_{KE}$) and the interaction (duration $\times$ category) on RPE; and a three-way, RM ANOVA was used to assess the effect of duration, condition, tissue site (VL, Gn, PFC) and the interactions on TSI. When

**Table 1** Speed, heart rate relative to individual maximum (%HR$_{MAX}$) and physiological cost index (PCI) during the three self-paced periods (SP1, SP2, SP3).

| | | Mean ± SD | 95% CI | % change | $p$ | $\eta_p^2$ |
|---|---|---|---|---|---|---|
| Speed (km h$^{-1}$) | SP1 | 10.3 ± 1.86 | 8.70 ± 11.8 | | 0.038[*] | 0.473[L] |
| | SP2 | 9.87 ± 1.96 | 8.24 ± 11.5 | −4.0 | | |
| | SP3 | 9.13 ± 2.07 | 7.40 ± 10.9 | −11.1 | | |
| %HR$_{MAX}$ (%) | SP1 | 78.5 ± 6.96 | 72.7 ± 84.3 | | 0.228 | 0.199[L] |
| | SP2 | 81.1 ± 6.52 | 75.7 ± 86.6 | 3.4 | | |
| | SP3 | 79.7 ± 6.22 | 74.5 ± 84.9 | 1.8 | | |
| PCI (bpm m$^{-1}$) | SP1 | 0.83 ± 0.19 | 0.68 ± 0.99 | | 0.009[*] | 0.608[L] |
| | SP2 | 0.90 ± 0.23 | 0.71 ± 1.09 | 7.8 | | |
| | SP3 | 0.97 ± 0.27 | 0.74 ± 1.19 | 15.6 | | |

**Notes.**

Data are group mean ± SD.

[*]significant main effect of duration, at $p < 0.05$.

[L]large effect.

ANOVAs indicated a significant difference, Fisher's LSD post-hoc test was used to conduct pairwise comparisons. The magnitude of changes was evaluated using partial eta-squared ($\eta_p^2$) interpreted according to Cohen's scale (*small* effect: $0.01 < \eta_p^2 < 0.06$; *moderate* effect: $0.06 < \eta_p^2 < 0.14$; *large* effect: $\eta_p^2 > 0.14$). Statistical analyses were performed using Statistica (version 13; StatSoft, Inc., Tulsa, OK, USA), and the level of significance was set at $p < 0.05$.

# RESULTS

All participants completed the 6TR, running a total distance of 58.3 ± 10.5 km (average speed: 9.7 ± 1.8 km h$^{-1}$) at an average %HR$_{MAX}$ of 79.2 ± 5.8% (HR: 139 ± 6 bpm). Body mass decreased 2.4 ± 0.1 kg following the 6TR (3.4 ± 1.8% of body weight). During each period of SP running, speed decreased and PCI increased significantly with large effect sizes, and %HR$_{MAX}$ did not change significantly (Table 1). There were significant and large increases in RPE as a function of duration ($p < 0.001$, $\eta_p^2 = 0.505$), with no significant effects of category ($p = 0.931$, $\eta_p^2 = 0.010$) or any interaction ($p = 0.559$, $\eta_p^2 = 0.111$) (Fig. 2).

The speeds used in each module were constant, by design, across modules in 10-CR, and were not significantly different across modules in HILL (M1: 8.79 ± 1.01 km h$^{-1}$, M2: 8.43 ± 1.47 km h$^{-1}$, M3: 8.13 ± 1.71 km h$^{-1}$, M4: 8.13 ± 1.71 km h$^{-1}$; $p = 0.083$, $\eta_p^2 = 0.267$). In the 10-CR condition, there were significant and large increases in %HR$_{MAX}$, $\dot{V}O_2$, OX$_{FAT}$ and $C_r$, significant and large decreases in RER, and no significant changes in $\dot{V}_E$, $C_r^J$, and $\dot{V}_E$:$\dot{V}O_2$ (Table 2). In the HILL condition, there were significant and large decreases in RER, significant and large increases in OX$_{FAT}$, and no significant changes in $\dot{V}O_2$, $\dot{V}_E$, $\dot{V}_E$:$\dot{V}O_2$, %HR$_{MAX}$, $C_r$ and $C_r^J$ (Table 3).

For TSI, there was a significant and large effect for site ($p = 0.001$, $\eta_p^2 = 0.611$), but there were no significant effects of duration ($p = 0.515$, $\eta_p^2 = 0.101$), condition ($p = 0.091$, $\eta_p^2 = 0.354$), or any interactions (Fig. 3). Post-hoc testing indicated that TSI in PFC was significantly greater than VL ($p = 0.001$) and Gn ($p = 0.001$), and VL and Gn were not

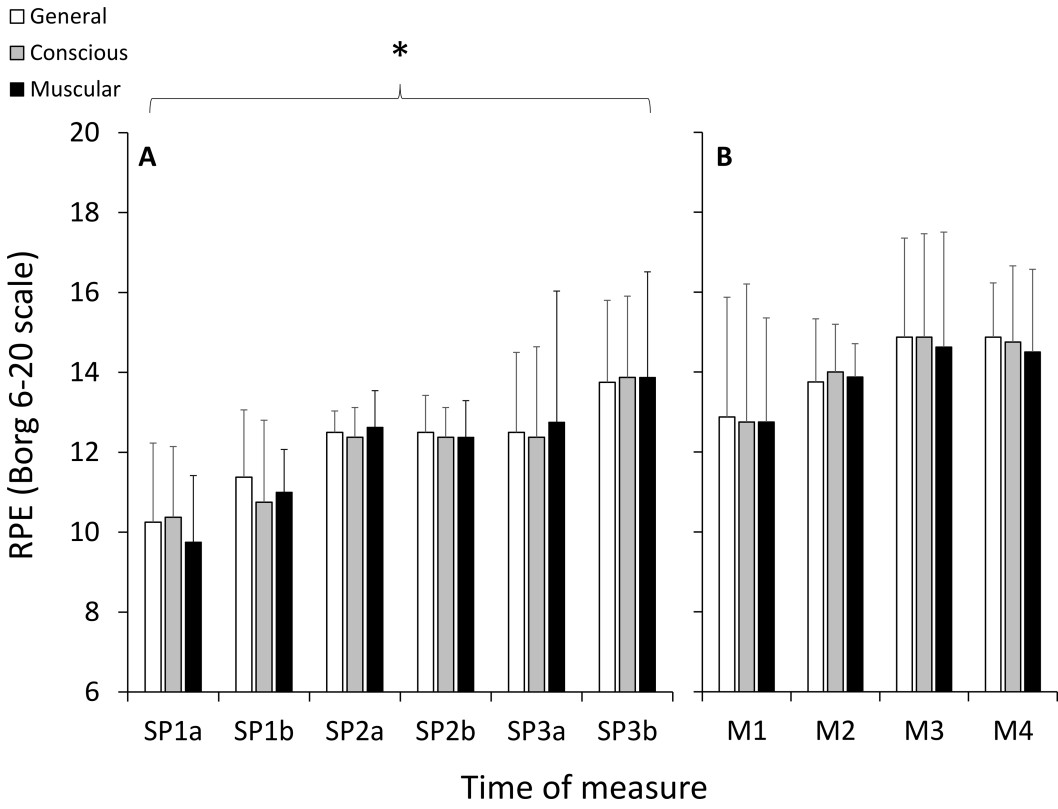

**Figure 2** Ratings of perceived exertion (RPE) measured using the 6–20 Borg scale during the self-paced periods of running **(A)**, and during the HILL condition during the modules **(B)**. *, significant effect of exercise duration; $p < 0.05$.

statistically different from each other ($p = 0.915$). For the RPE variables during the HILL condition, there were no significant effects of duration ($p = 0.164$, $\eta_p^2 = 0.212$), category ($p = 0.906$, $\eta_p^2 = 0.014$) or any interactions ($p = 0.986$, $\eta_p^2 = 0.023$) (Fig. 2).

## DISCUSSION

The aim of this study was to investigate the effect of exercise duration on cardiorespiratory responses ($\dot{V}O_2$, $\dot{V}_E$, $\dot{V}_E$:$\dot{V}O_2$, RER, %HR$_{MAX}$), energy cost of running ($C_r$ and $C_r^J$), muscle tissue and cerebral oxygenation (TSI), and differentiated perceived exertion (RPE), at moderate and heavy exercise intensities within a 6 h treadmill run. Participants completed an average of 58.3 km, and there was a progressive decrease in running speed throughout the 6TR (positive pacing) with increases in PCI (cardiac drift) and RPE as a function of duration during the self-paced periods of running. During 10-CR, RER and running economy decreased, and %HR$_{MAX}$, OX$_{FAT}$ and $C_r$ increased. During HILL, RER decreased, and OX$_{FAT}$ increased.

The 10-CR components of the modules were performed on level gradient (1%) and at a speed comparable to the average speed of the 6TR. As expected, exercise intensity during 10-CR was moderate (53.9–56.4% of $\dot{V}O_{2MAX}$). We measured increases in cardiorespiratory parameters %HR$_{MAX}$ and $\dot{V}O_2$, as well as changes in substrate utilization as estimated from

**Table 2** Changes in the cardiorespiratory variables and energy cost, measured during the 10-CR condition of the modules.

| | Module | Mean $\pm$ SD | 95% CI | % change | $p$ | $\eta_p^2$ |
|---|---|---|---|---|---|---|
| $\dot{V}O_2$ (mL kg$^{-1}$ min$^{-1}$) | 1 | $32.1 \pm 3.3$ | $29.4 \pm 34.9$ | | | |
| | 2 | $31.2 \pm 2.6$ | $29.0 \pm 33.4$ | $-2.6$ | $0.020^*$ | $0.367^L$ |
| | 3 | $32.7 \pm 3.1$ | $30.2 \pm 35.3$ | $+2.2$ | | |
| | 4 | $33.3 \pm 3.3$ | $30.5 \pm 36.0$ | $+3.6$ | | |
| $\dot{V}_E$ (L min$^{-1}$) | 1 | $49.2 \pm 6.47$ | $43.8 \pm 54.6$ | | | |
| | 2 | $50.0 \pm 6.45$ | $44.6 \pm 55.4$ | $+2.0$ | $0.061$ | $0.290^L$ |
| | 3 | $52.6 \pm 7.63$ | $46.3 \pm 59.0$ | $+7.1$ | | |
| | 4 | $53.8 \pm 6.17$ | $48.6 \pm 58.9$ | $+10.0$ | | |
| $\dot{V}_E{:}\dot{V}O_2$ | 1 | $25.8 \pm 1.5$ | $24.5 \pm 27.0$ | | | |
| | 2 | $27.0 \pm 3.0$ | $24.6 \pm 29.5$ | $+5.0$ | $0.218$ | $0.187^L$ |
| | 3 | $27.1 \pm 3.3$ | $24.3 \pm 29.8$ | $+4.9$ | | |
| | 4 | $27.5 \pm 3.1$ | $24.9 \pm 30.0$ | $+6.6$ | | |
| RER | 1 | $0.92 \pm 0.06$ | $0.87 \pm 0.98$ | | | |
| | 2 | $0.89 \pm 0.05$ | $0.85 \pm 0.93$ | $-3.8$ | $0.023^*$ | $0.359^L$ |
| | 3 | $0.87 \pm 0.04$ | $0.84 \pm 0.90$ | $-5.7$ | | |
| | 4 | $0.86 \pm 0.03$ | $0.84 \pm 0.89$ | $-6.3$ | | |
| $OX_{FAT}$ (g min$^{-1}$) | 1 | $0.29 \pm 0.23$ | $0.13 \pm 0.45$ | | | |
| | 2 | $0.41 \pm 0.16$ | $0.30 \pm 0.52$ | $+71$ | $0.030^*$ | $0.342^L$ |
| | 3 | $0.50 \pm 0.12$ | $0.42 \pm 0.58$ | $+106$ | | |
| | 4 | $0.52 \pm 0.14$ | $0.43 \pm 0.62$ | $+114$ | | |
| $C_r$ (mL kg$^{-1}$ m$^{-1}$) | 1 | $0.172 \pm 0.020$ | $0.158 \pm 0.185$ | | | |
| | 2 | $0.166 \pm 0.016$ | $0.155 \pm 0.177$ | $-3.4$ | $0.004^*$ | $0.464^L$ |
| | 3 | $0.175 \pm 0.019$ | $0.163 \pm 0.188$ | $+2.1$ | | |
| | 4 | $0.179 \pm 0.020$ | $0.165 \pm 0.192$ | $+3.8$ | | |
| $C_r^J$ (J kg$^{-1}$ m$^{-1}$) | 1 | $3.56 \pm 0.40$ | $3.23 \pm 3.89$ | | | |
| | 2 | $3.41 \pm 0.34$ | $3.13 \pm 3.70$ | $-3.7$ | $0.062$ | $0.289^L$ |
| | 3 | $3.59 \pm 0.40$ | $3.26 \pm 3.92$ | $+1.1$ | | |
| | 4 | $3.64 \pm 0.40$ | $3.31 \pm 3.98$ | $+2.6$ | | |
| %HR$_{MAX}$ (%) | 1 | $82.5 \pm 8.3$ | $75.5 \pm 89.5$ | | | |
| | 2 | $86.4 \pm 8.1$ | $79.6 \pm 93.2$ | $+4.8$ | $0.001^*$ | $0.817^L$ |
| | 3 | $88.6 \pm 7.3$ | $82.5 \pm 94.7$ | $+7.6$ | | |
| | 4 | $89.5 \pm 6.7$ | $81.1 \pm 97.8$ | $+13.6$ | | |

**Notes.**

Data are group mean $\pm$ SD.

*significant main effect of duration, at $p < 0.05$.

$^L$large effect.

$\dot{V}O_2$, Oxygen utilization; $\dot{V}_E$, Minute ventilation; $\dot{V}_E{:}\dot{V}O_2$, Ventilatory equivalent; RER, Respiratory Exchange Ratio; $C_r$ and $C_r^J$, Energy cost of running; %HR$_{MAX}$, Heart rate relative to individual maximum.

changes in $OX_{FAT}$ and RER. These results were expected, as previous studies conducted following events ranging 10–149 km, performed at intensities ranging 39–85% of $\dot{V}O_{2MAX}$ (including self-paced events), have consistently reported increases in $\dot{V}O_2$ (*Gimenez et al., 2013*; *Hausswirth et al., 1996*; *Hunter & Smith, 2007*; *Kyröläinen et al., 2000*; *Nicol, Komi & Marconnet, 1991*; *Xu & Montgomery, 1995*) and decreases in RER (*Gimenez et al., 2013*;

**Table 3** Changes in the cardiorespiratory variables and energy cost of running, measured during the HILL condition of the modules.

| | Module | Mean ± SD | 95% CI | % change | $p$-value | $\eta_p^2$ |
|---|---|---|---|---|---|---|
| $\dot{V}O_2$ (mL kg$^{-1}$ min$^{-1}$) | 1 | 47.4 ± 6.7 | 41.8 ± 52.9 | | 0.067 | 0.283[L] |
| | 2 | 43.9 ± 9.8 | 37.0 ± 50.9 | −7.5 | | |
| | 3 | 42.9 ± 9.8 | 34.6 ± 51.1 | −9.9 | | |
| | 4 | 42.5 ± 11.5 | 33.0 ± 52.1 | −10.9 | | |
| $\dot{V}_E$ (L min$^{-1}$) | 1 | 80.1 ± 14.2 | 68.2 ± 92.0 | | 0.214 | 0.188[L] |
| | 2 | 73.3 ± 14.9 | 60.9 ± 85.7 | −8.0 | | |
| | 3 | 73.4 ± 14.1 | 61.6 ± 85.2 | −7.6 | | |
| | 4 | 73.0 ± 17.7 | 58.2 ± 87.9 | −8.1 | | |
| $\dot{V}_E{:}\dot{V}O_2$ | 1 | 24.0 ± 2.0 | 22.3 ± 25.7 | | 0.382 | 0.133[M] |
| | 2 | 23.9 ± 2.6 | 21.7 ± 26.1 | −0.4 | | |
| | 3 | 24.9 ± 3.3 | 22.1 ± 27.6 | +3.4 | | |
| | 4 | 24.8 ± 2.0 | 23.1 ± 26.5 | +3.7 | | |
| RER | 1 | 1.01 ± 0.07 | 0.95 ± 1.06 | | 0.013[*] | 0.396[L] |
| | 2 | 0.96 ± 0.08 | 0.89 ± 1.02 | −4.9 | | |
| | 3 | 0.96 ± 0.04 | 0.92 ± 0.99 | −4.5 | | |
| | 4 | 0.94 ± 0.04 | 0.91 ± 0.98 | −5.8 | | |
| $OX_{FAT}$ (g min$^{-1}$) | 1 | 0.13 ± 0.15 | 0.03 ± 0.24 | | 0.018[*] | 0.419[L] |
| | 2 | 0.21 ± 0.15 | 0.10 ± 0.32 | +62 | | |
| | 3 | 0.18 ± 0.13 | 0.08 ± 0.27 | +31 | | |
| | 4 | 0.17 ± 0.13 | 0.07 ± 0.26 | +20 | | |
| $C_r$ (mL kg$^{-1}$ m$^{-1}$) | 1 | 0.299 ± 0.023 | 0.283 ± 0.315 | | 0.403 | 0.127[L] |
| | 2 | 0.287 ± 0.027 | 0.268 ± 0.306 | −4.0 | | |
| | 3 | 0.289 ± 0.032 | 0.267 ± 0.311 | −3.3 | | |
| | 4 | 0.284 ± 0.034 | 0.260 ± 0.308 | −5.1 | | |
| $C_r^J$ (J kg$^{-1}$ m$^{-1}$) | 1 | 6.31 ± 0.48 | 5.90 ± 6.71 | | 0.206 | 0.192[L] |
| | 2 | 5.99 ± 0.55 | 5.54 ± 6.45 | −4.9 | | |
| | 3 | 6.04 ± 0.69 | 5.47 ± 6.62 | −4.2 | | |
| | 4 | 5.92 ± 0.76 | 5.29 ± 6.55 | −6.0 | | |
| $\%HR_{MAX}$ (%) | 1 | 97.8 ± 5.96 | 92.8 ± 102.8 | | 0.164 | 0.337[L] |
| | 2 | 98.8 ± 3.88 | 95.6 ± 102.1 | +1.2 | | |
| | 3 | 99.2 ± 3.74 | 96.1 ± 102.4 | +1.8 | | |
| | 4 | 100.4 ± 3.82 | 95.7 ± 105.2 | +3.3 | | |

**Notes.**

Data are group mean ± SD.

[*]significant main effect of duration, at $p < 0.05$.

[M]moderate effect.

[L]large effect.

$\dot{V}O_2$, Oxygen utilization; $\dot{V}_E$, Minute ventilation; $\dot{V}_E{:}\dot{V}O_2$, Ventilatory equivalent; RER, Respiratory Exchange Ratio; $C_r$ and $C_r^J$, Energy cost of running; $\%HR_{MAX}$, Heart rate relative to individual maximum.

*Millet et al., 2000*; *Vernillo et al., 2016b*; *Vernillo et al., 2015*). In the current study, the increased $C_r$ without changes in $C_r^J$ are likely explained by changes in substrate utilization based on in $OX_{FAT}$ and RER. These results extends findings of increased $C_r$ after a marathon (*Brueckner et al., 1991*; *Hausswirth et al., 1996*; *Hunter & Smith, 2007*; *Kyröläinen et al., 2000*; *Nicol, Komi & Marconnet, 1991*; *Xu & Montgomery, 1995*), as well as the absence of

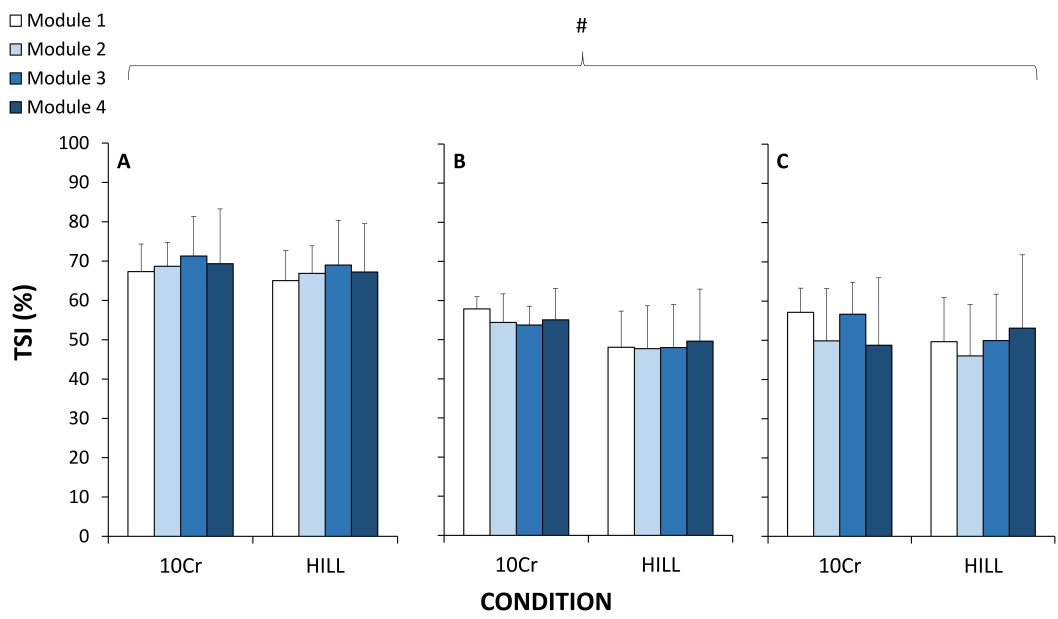

**Figure 3** Tissue saturation index (TSI) measured with near-infrared spectroscopy at the vastus lateralis muscle (A), gastrocnemius muscle (B) and prefrontal cortex (C). #, significant effect of tissue site; $p < 0.05$.

changes in $C_r^J$ after a 65-km mountain ultramarathon (*Vernillo et al., 2015*). As such, the different outcomes in studies reporting changes in the energy cost of running could have originated from the use of different methods of calculation ($C_r$ and $C_r^J$).

The HILL components of the modules were performed on an uphill gradient (10%) and at a heavy exercise intensity (71.9–79.7% of $\dot{V}O_{2MAX}$). During HILL, we measured only changes in substrate utilization, with decreased RER and increased $OX_{FAT}$, with non-significant decreases in $\dot{V}O_2$. These results were not expected, but indicate that the ability to exercise at a heavy intensity (*Gaesser & Poole, 1996*) may not be altered within the 6TR. Additionally, the absence of change in $\dot{V}_E$ and $\dot{V}_E:\dot{V}O_2$ at moderate and heavy exercise intensities contrasts with other studies following trail ultramarathons (*Millet et al., 2000*; *Vernillo et al., 2014*), and indicates that prolonged running with bouts of heavy intensity, but no downhill component, has small effects on the cardiorespiratory response of trained ultramarathon runners. The absence of change in $C_r^J$ in our study is also in agreement with results of a study following a 65-km mountain ultramarathon performed on trails (*Vernillo et al., 2015*), but differs from other studies having measured improved (decreased) cost of running and walking following longer (330 km) mountain ultramarathons (*Vernillo et al., 2016a*; *Vernillo et al., 2014*). However, in the current study the uphill gradient condition was chosen to increase exercise intensity while minimizing the contribution of elastic energy in gait, in order to control the possibility raised by other authors that changes in microvascular structure could have increased muscle tissue energy requirements (*Vernillo et al., 2016a*). Another perspective to explain the absence of change in $C_r^J$ may reside in the specific muscular adaptations derived from the typical high mileage training performed by ultramarathon runners. Unfortunately, we could not test

this hypothesis, as we did not collect specific training background variables in the current study participants. Nonetheless, the results of 10-CR and HILL indicate that the effects of prolonged running with no downhill gradient were associated with changes in substrate utilization, increased cardiorespiratory parameters and Cr only at the moderate intensity, and with unchanged $\dot{V}_E$ and $\dot{V}_E:\dot{V}O_2$.

Therefore, another major finding of the current study is the absence of change of tissue oxygenation in two muscle and one cerebral sites as a function of exercise duration, across either the 10-CR and HILL components of the modules. We report differences across sites, where TSI in PFC was higher than both VL and Gn. Despite the expected and marked differences in exercise intensity, TSI also did not decrease significantly between the 10-CR and HILL conditions. Although one previous study observed tissue oxygenation alterations following prolonged duration running (*Vernillo et al., 2016a*), it is possible that the important downhill running component could have contributed to impair tissue structure and microcirculatory function, leading to altered tissue oxygenation (*Vernillo et al., 2016a*). In the current study, we sought to minimize the potentially confounding effects of the mode of muscle contraction (such as elicited by repeated downhill running). Further, as in the current study, differences in site, but no effect of exercise duration, were reported for muscle and cerebral oxygenation during a 4 h cycling exercise (*Rupp et al., 2013*). However, alterations in oxygen-dependent metabolic processes attributed to decreased mitochondrial efficiency, determined using biopsies of the VL muscle, were measured after a 24 h simulated adventure race (running, kayaking, cycling) at ∼60% of $\dot{V}O_{2MAX}$ (*Fernström et al., 2007*). As such, it is possible that the exercise duration was too short in the current study to elicit metabolic changes at tissue level. Therefore, the results of the current study indicate that tissue oxygenation in the microvasculature of two muscles and one cerebral sites was not impaired by exercise duration at moderate and heavy exercise intensities.

Finally, we report an increase in RPE in three categories used in this study as a function of duration across SP, but the categories of RPE were not different to each other. The absence of differences between RPE categories, and the relatively low group mean values are similar to those reported during other prolonged running events (*Kerhervé, Millet & Solomon, 2015*; *Vercruyssen et al., 2012*). Therefore, despite increased RPE values as a function of duration, the absence of differences across RPE categories indicate that differentiated RPE scales potentially do not provide a useful measure to differentiate the specific categories used in the current study for prolonged duration exercise. Consequently, we can hypothesize that RPE is a general indicator of psycho-physiological load, influenced by fatigue (*Marcora, Bosio & Moree, 2008*), but independent from sensory afferents (*Marcora, 2009*). Trained endurance athletes could also readily use, knowingly or not, specific strategies to limit or reduce RPE during fatiguing endurance exercise, such as self-talk (*Blanchfield et al., 2014*). Therefore, additional research is required to investigate the etiology of psycho-physiological fatigue, including the role of attentional focus, cognitive control and metacognition in relation to indicators of metabolic load and efficiency. Combined, the findings of the current study indicate that at moderate intensity, increased exercise duration was associated with increased cardiorespiratory load, increased energy cost of running only when expressed as oxygen consumption per unit distance, changes in substrate utilization, and increased

perceived exertion. During periods of heavy exercise intensity, increased exercise duration was associated with changes in substrate utilization but not cardiorespiratory load. Overall, exercise duration was not associated with changes in ventilatory efficiency, energy cost of running expressed as energy expenditure per unit distance, or tissue oxygenation, regardless of exercise intensity.

There were three main limitations to this study. First, we could not perform continuous measurements of TSI because the devices could not remain in contact with the skin for extended periods without causing discomfort and additional sweating, the latter being incompatible with consistent measurements. Two trained investigators were present to fix and remove the devices from the testing sites at each module to reduce the total time stopped (18 $\pm$ 1 min), and we minimized the effect of the time stopped by adding periods of self-paced running before and after the 10-CR and HILL conditions. There were also non-significant changes in speed in the HILL condition, which could have minimized expected increases in metabolic load. However, we permitted to vary speed in this condition to ensure the completion of the 6TR, and we aimed to establish whether exercise performed at heavy intensity would alter cardiorespiratory efficiency, energy cost of running and tissue oxygenation compared to self-paced running. Second, we did not record weekly mileage or lifetime training experience, which could constitute important explanatory factors for the variables collected. Third, we did not record nutrition or hydration intake, although these could have important implications for the results of the current study. This constitutes an important direction for future studies.

## CONCLUSIONS

Speed decreased overall, and heart rate (cardiac drift) and the subjective perception of effort increased, during a 6 h treadmill run. Running at $\sim$54–56% $\dot{V}O_{2MAX}$ on level ground was associated with increases in systemic oxygen consumption, heart rate and decreases in RER, and running at $\sim$72–82% $\dot{V}O_{2MAX}$ was associated only with decreases in RER. There were no changes in ventilatory efficiency, energy cost of running, and muscular and cerebral tissue oxygenation as a function of exercise duration, at moderate and heavy exercise intensity. Therefore, the major changes observed within a 6 h treadmill run with no downhill component were a modification of substrate utilization and increased heart rate, but exercise capacity remained unchanged at moderate and heavy exercise intensities. We recommend future research to be directed to establishing the potential links between the dynamics of energy cost of running and nutrition and hydration status, as well as training status and lifetime training experience.

## ACKNOWLEDGEMENTS

The authors would like to thank all the participants for their effort and valuable time, the university's laboratory technicians (Stephen Bishop, Alysha Hyde, Darren Morrow and Ava Kerr) for their technical support, and Ms. Benjie Bartos for editing and correcting the manuscript.

### Funding

The authors received no specific funding for this work. The funders had no role in study design, data collection and analysis, decision to publish, or preparation of the manuscript.

### Competing Interests

The authors declare there are no competing interests.

### Author Contributions

- Hugo A. Kerhervé and Colin Solomon conceived and designed the experiments, performed the experiments, analyzed the data, wrote the paper, prepared figures and/or tables, reviewed drafts of the paper.
- Scott McLean, Karen Birkenhead and David Parr performed the experiments, wrote the paper, prepared figures and/or tables, reviewed drafts of the paper.

### Human Ethics

The following information was supplied relating to ethical approvals (i.e., approving body and any reference numbers):

This study was approved by the Human Research Ethics Committee of the University of the Sunshine Coast (Approval Number: S/12/432).

### Data Availability

Kerherve, Hugo; McLean, Scott; Birkenhead, Karen; Parr, David; Solomon, Colin (2017): Influence of exercise duration on cardiorespiratory responses, energy cost and tissue oxygenation within a 6 hour treadmill run. figshare.

https://doi.org/10.6084/m9.figshare.4822093.v1.

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
