# Peer review of "Influence of exercise duration on cardiorespiratory responses, energy cost and tissue oxygenation within a 6 hour treadmill run"

_PeerJ, doi:10.7717/peerj.3694_

## Round 0.1 · original submission · Minor Revisions

Dear Hugo, upon resubmission, please address all of the reviewers' comments and concerns. Please highlight the changes you have made in the manuscript, and also provide a response to each comment. Where you have chosen to refute any of the suggestions by the reviewers, please provide a detailed rationale. I look forward to the revised version.

Scotty

·

Basic reporting

Abstract:
Please insert the correct VO2max units throughout the manuscript: "mL•min•kg-1" change to "mL•kg-1•min-1"

Since you did not directly measure beta-oxidation. I would change to a reduced RER. I would suggest the authors estimate fat oxidation with the use of RER, and Oxygen Utilization.

If you observed an increase in VO2 at the 10 km/h running speed, this would indicate a reduced running economy, despite no change in the energy cost of running.

Introduction:
Line 52. "It has been suggested increases".... Change to "It has been suggested that increases..."

Did you record years of training experience? Or average weekly training time? These are often more informative then VO2max values.

Was nutrition monitored or recorded? I know it was not controlled. But nutrition could have had a large effect on the results.

Line 156-157. RER is VCO2:VO2. You have it reversed? Please correct.

I am not clear how you obtained J•kg•m-1 . Can you please show me how you got these units?

It is unclear why the two-way RM ANOVA and three-way RM ANOVA's were used. These were not clear from the aim of the study.

Experimental design

I completely understand the small sample size, but was a sample size calculation done prior to the study?

Validity of the findings

Was the metabolic system calibrated before and after? What is the reliability of the metabolic system in the laboratory used?

Additional comments

The study is very unique, with a 6 hours of treadmill running. The authors point out on several occasions that no downhill running was included as this may alter the findings. However, what is the external validity of such findings (ie. are there any ultramarathon courses with no downhill sections)?

I also feel that is important to point out that running economy was reduced despite no change in the energy cost of running.

Lastly, I suggest the authors calculate fat oxidation instead of assuming a reduction in RER is equivalent to an increase in beta-oxidation.

Reviewer 2 ·

Basic reporting

no comment

Experimental design

no comment

Validity of the findings

Speculation is welcome, but should be identified as such.

Additional comments

General comments: The current paper is relatively well-written and the topic related to endurance exercise and including many laboratory-running measurements is quite interesting. However, the paper requires some important clarifications, especially during the methods and discussion parts. To improve the global reading of the present manuscript, it would be judicious to use more abreviations throughout the text, such as 10-CR or HILL from the beginning to the end of proposed statements. The authors are requested to improve the quality of their work by taking account into the specific comments described below.



INTRODUCTION:

. L46: suggestion: Maximal oxygen uptake (VO2max), the fractional rate of VO2max (%VO2MAX) and the energy cost of running (Cr), i.e. the submaximal metabolic demand (VO2) per unit of distance covered are…
. L48: Insert “may be” instead of “are” affected
. L52: Insert “that” after suggested.
. L53-56 : The reviewer is in agreement with the statements used by the authors but the sentence needs to be re-written by presenting firstly the effects of exercise duration until 8-h on VO2 and secondly, the effects of ultramarathon on the same variable.
. L56-61 : Could you reformulate these sentences please.

METHODS :
Could you extend or precise the weekly running mileage for your experimental group. Does your population cycle during their training sessions ?
L111: Replace maximum VO2 by VO2max.
L132 : Why did you select a +10% slope ?
L133 : For HILL, is the speed corresponding to 70%VO2max or the highest possible speed as you reported in the manuscript ? clarify the confusion please.
L160 : It would have also been interesting to convert J.kg-1.m-1 in J.m-1 shunting the body mass which may decrease with exercise duration and exert an additional influence on final CrJ values. What ‘s your position about this choice of unit.
L176 : What was the average time to attach and remove the NIRS devices during the four modules ? Could you provide more details about the position of the NIRS device on the prefrontal cortex.

RESULTS : well presented.

DISCUSSION : in the paragraph 17, I suggest that you also present the results including oxygenation and metabolic responses, not only PCI and RPE.
L245-246 ; precise the two exercise intensities.
L250- : Could you reformulate this sentence (i.e. repeated terms) and by including RER values to characterize the beta-oxidation.
L253 : Could you reformulate « at intensities < 80% of maximum ».
L257 : Beta-oxidation is not directly measured in the current work but extrapolated from RER values. It should be more appropriate to use the RER decrease firstly before inferring on beta-oxidation.
L263 : replace « mild » by small.
In the paragraph 19, the authors discussed their results focusing on the lack of CrJ alteration with exercise duration. Some hypotheses are proposed but the training status of the experimental group has not discussed in this part. The high training background and thereby, muscular adaptations in ultramarathon runners might constitute an additional hypothesis to explain the lack of CrJ alteration with exercise duration, either on level ground or HILL condition. Could you explore this type of hypothesis.
You observed an absence of CrJ alteration on level ground, but at a moderate intensity (10 km/h) or at a self-selected speed or both. Could you precise in the discussion part what « level ground » means to clarify the reading. Furthermore, for the self-selected speed-based period of measurement, you observed a decrease in running speed across the modules and this decrease in speed may explain why CrJ did not change across the time in the case where VO2 continuously increase. You have not discussed the ratio between change in VO2 and change in speed to explain CrJ responses across the time.
L283-286 : could you reformulate this sentence.
L290-295 : The authors hypothesized that it is likely that the exercise duration during the HILL and 10-CR modules was too short to induce any significant changes in TSI responses throughout muscular sites. A link with a 24-h outdoor exercise using biopsies methods has been used, but it would have been better to create a link with studies investigating the evolution of NIRS parameters with exercise duration. Furthermore, it would be interesting to present limitations of NIRS devices in dynamic exercices in an attempt to explain the lack of TSI changes with exercise duration.
L312 : I am a bit confused when the authors reported that the 6TR did affect the energy cost of running.
Conclusions : insert future research directions at the end of conclusion part.

Title table 1. Heart rate is mentioned twice. Change heart rate relative to individual maximum. What is the signification of the statistical symbol used ? Which difference ? Could you apply these modifications to table 2 and table 3.
Figure 1. Could you insert the abreviations 10-CR and HILL, moderate and heavy exercises in the figure content
In the results section, you wrote that TSI responses were significantly higher in PFC compared to VL and GL, but when observed the figure 3 and especially B and C, the reviewer has the impression that no difference exist between sites. Given that this impression is purely visual, could you confirm the statistical inference of your results. I suggest that you also change the statistical representation, not really clear.








.

Reviewer 3 ·

Basic reporting

-This is a nicely written manuscript that is clear in its rationale, methods, analyses, and conclusions. No major concerns.
- No notable grammatical issues.
- Data is nicely presented.

Experimental design

- The design is appropriate for the research question.
- Hypotheses are appropriate and rationalized.
- Methods are nicely described - no concerns.

Validity of the findings

- The findings are interesting and worth publishing. I'd suggest linking the data with work showing the effects of prior heavy exercise on moderate intensity exercise (references noted below).
- Statistics are nicely described.

Additional comments

- Please define "reverse cardiovascular drift".
- Would suggest inclusion of some priming literature on the amplitude of VO2 -- as these studies provide some physiologic insight into the mechanisms of how prior heavy exercise influences subsequent moderate intensity exercise (DiMenna et al. J Appl Physiol 2009 107(6):1743-56; Wilkerson et al. J Appl Physiol. 2004 97(4):1227-36; Gurd et al. J Appl Physiol. 2005 98(4):1371-8). These seminal papers highlight important physiology about priming effects on VO2.
-

---

## Round 0.2 · accepted · Accept

Thanks again for your submission and congratulations on the acceptance. Upon final review, there are a few typographical changes suggested by reviewer 1 that should be attended to.

Reviewer 2 ·

Basic reporting

no comment

Experimental design

clear

Validity of the findings

clear

Additional comments

Responses and/or modifications from the authors are considerably improved the manuscript in terms of methodological quality, results and discussion presentation. I recommend the publication of the current work in the Running scope, when applying the minor specific comments below.

L50: ….that increases in Cr correlate positively with….

L54: remove “s” : afetr a 65-km trail ultramarathon

L54: suggestion : Similarly, following a 65 km mountain ultramarathon….

L132: I suggest authors to report why the +10% slope was selected in this protocol, on the basis of feedback comments.

Paragraph number 7: I suggest authors to remove “complete” (repeated twice) in the following sentence…. to complete the condition and/or (HERE) the 6TR,....

Reviewer 3 ·

Basic reporting

The authors have addressed my comments.

Experimental design

The authors have addressed my comments.

Validity of the findings

The authors have addressed my comments.

Additional comments

The authors have addressed my comments.